# Exercise Intensity in Patients with Cardiovascular Diseases: Systematic Review with Meta-Analysis

**DOI:** 10.3390/ijerph18073574

**Published:** 2021-03-30

**Authors:** Catarina Gonçalves, Armando Raimundo, Ana Abreu, Jorge Bravo

**Affiliations:** 1Departamento de Desporto e Saúde, Escola de Saúde e Desenvolvimento Humano, Universidade de Évora, Largo dos Colegiais, 7000 Évora, Portugal; cjg@uevora.pt (C.G.); ammr@uevora.pt (A.R.); 2Comprehensive Health Research Centre (CHRC), Universidade de Évora, Largo dos Colegiais, 7000 Évora, Portugal; 3Servico de Cardiologia, Hospital Universitário de Santa Maria/Centro Hospitalar Universitário Lisboa Norte (CHULN), Centro Académico de Medicina de Lisboa (CAML), Centro Cardiovascular da Universidade de Lisboa (CCUL), Faculdade de Medicina, Universidade de Lisboa, 1649-028 Lisboa, Portugal; ananabreu@hotmail.com

**Keywords:** cardiac rehabilitation, cardiorespiratory fitness, exercise therapy, heart diseases, high-intensity intermittent exercise

## Abstract

Exercise-induced improvements in the VO_2_peak of cardiac rehabilitation participants are well documented. However, optimal exercise intensity remains doubtful. This study aimed to identify the optimal exercise intensity and program length to improve VO_2_peak in patients with cardiovascular diseases (CVDs) following cardiac rehabilitation. Randomized controlled trials (RCTs) included a control group and at least one exercise group. RCTs assessed cardiorespiratory fitness (CRF) changes resulting from exercise interventions and reported exercise intensity, risk ratio, and confidence intervals (CIs). The primary outcome was CRF (VO_2_peak or VO_2_ at anaerobic threshold). Two hundred and twenty-one studies were found from the initial search (CENTRAL, MEDLINE, CINAHL and SPORTDiscus). Following inclusion criteria, 16 RCTs were considered. Meta-regression analyses revealed that VO_2_peak significantly increased in all intensity categories. Moderate-intensity interventions were associated with a moderate increase in relative VO_2_peak (SMD = 0.71 mL-kg^−1^-min^−1^; 95% CI = [0.27–1.15]; *p* = 0.001) with moderate heterogeneity (I^2^ = 45%). Moderate-to-vigorous-intensity and vigorous-intensity interventions were associated with a large increase in relative VO_2_peak (SMD = 1.84 mL-kg^−1^-min^−1^; 95% CI = [1.18–2.50], *p* < 0.001 and SMD = 1.80 mL-kg^−1^-min^−1^; 95% CI = [0.82–2.78] *p* = 0.001, respectively), and were also highly heterogeneous with I^2^ values of 91% and 95% (*p* < 0.001), respectively. Moderate-to-vigorous and vigorous-intensity interventions, conducted for 6–12 weeks, were more effective at improving CVD patients’ CRF.

## 1. Introduction

Cardiovascular diseases (CVDs) are the leading cause of mortality in today’s society, being responsible for up to one-third of all deaths worldwide and 50% of all deaths in Europe, and this scenario is expected to worsen in the coming years [1].

The concept of cardiac rehabilitation (CR) has been defined as the effort towards cardiovascular risk factor reduction, designed to lessen the chance of a subsequent cardiac event, and to slow and perhaps stop the progression of the disease process. In the context of CR programs, exercise training has been recognized as one of the main components, combined with education, control, pharmacological adherence and lifestyle changes of cardiovascular risk factors [2]. Physical exercise inclusion in CR programs resulted in several beneficial effects on cardiovascular functional capacity, quality of life, risk factor modification, psychological profile, hospital readmissions, and mortality [3,4]. Such benefits can be justified by a 20% reduction in mortality from all causes and in the levels of cardiorespiratory fitness (CRF) for each metabolic equivalent improvement (MET) in CRF of patients with CVD [5].

Exercise programs for patients with CVD traditionally involve mostly low- to moderate-intensity continuous aerobic exercise training, with the consensus that one of the benefits of aerobic exercise is the increase in peak oxygen uptake (VO_2_peak) [6,7,8]. Continuous aerobic exercise training implicates higher durations under moderate-intensity and nonvariable aerobic activity (60–80% of VO_2_peak) [9,10,11,12], compared to high-intensity protocols, which consist of intermittent, short high-intensity work periods (85–100% of VO_2_peak) with relative resting periods [13,14].

Exercise intensity appears to influence the number of cardioprotective benefits achieved from aerobic exercise [15,16]. The current consensus recommends that exercise intensity prescribed for patients with CVD should be approximately 60% of the maximal heart rate (MHR), 50% of the heart rate reserve (HRR), or 12–13 on the Borg scale. Intensities around 85% MHR, 80% HRR, or 15–16 on the Borg scale should represent the upper limits [6]. Additionally, high-intensity protocols (85–100% of VO_2_peak) appear to be of particular interest to scientists, considering their application in patients with CVD based on the effects on the cardiorespiratory and muscle systems [7]. High-intensity protocols elicit a greater training stimulus than moderate continuous exercise in improving maximal aerobic capacity [8,9,10,11,12,13,14,15,16,17,18,19]. In addition, high-intensity exercise appears to improve the limiting factors of VO_2_peak, and VO_2_peak itself has been found to be more effective in improving cardiovascular risk factors than moderate-intensity exercise [17,19].

Training sessions based on moderate-intensity continuous exercise have shown improvements in HRR after eight weeks [20] and after 12 weeks [21,22]. Moderate- to high-intensity continuous exercise (6 and 12 MET, corresponding to 21 and 42 mL-kg^−1^-min^−1^ of VO_2_peak) has also been shown to reduce all-cause mortality in healthy individuals, independent of activity duration [7], and reduce the risk of heart disease [15], supporting the need to further investigate the potential health effects of protocols based on higher intensities. Therefore, during the last two decades, several studies have demonstrated that high-intensity exercise protocols induce more beneficial cardiovascular adaptations in patients with mild-to-severe heart disease when compared to moderate-intensity exercise protocols [8,17,18,19].

A recent meta-analysis [23] reported higher improvements in maximal aerobic capacity after high-intensity interval training (HIIT) programs compared to moderate-intensity programs. Nevertheless, the optimum exercise intensity prescription in patients with CVD is still a subject of debate. A recent systematic review on the topic [24] did not report optimal intensity prescription (e.g., the intensity interval that is most effective during exercise interventions to induce favorable changes in aerobic capacity). Thus, despite the literature being replete with studies showing that regular and structured exercise is beneficial for CVD patients, the optimal intensity and length of exercise interventions that bring about greater benefits remain equivocal. Hence, the objective of this systematic review with meta-analysis was to identify, through Randomized Controlled Trials (RCTs) of exercise-based CR, the most effective exercise intensity and intervention length to optimize VO_2_peak in patients with CVD.

## 2. Materials and Methods

This systematic review was undertaken as detailed in the protocol registered with PROSPERO (Registration Number CRD42018097319).

### 2.1. Search Strategy

The search strategies were designed in accordance with the methods suggested by the Cochrane Handbook for Systematic Reviews of Interventions [25]. The following databases were searched from their inception to January 2021: Cochrane Central Register of Controlled Trials (CENTRAL), MEDLINE (Ovid), CINAHL (EBSCO) and SPORTDiscus. Data are provided as the risk difference (95% CI), based on RCTs published until January 2021, ensuring that all studies have been included if reporting data on established outcomes. Reference lists of eligible studies were also systematically searched.

We used the PICO model [26] to identify free text terms and controlled vocabulary terms to create our searches. The following key concepts were chosen: ‘‘Patients with cardiovascular diseases’’ AND ‘‘Cardiac Rehabilitation’’ AND ‘‘Exercise Intensity’’ AND ‘‘Cardiorespiratory Fitness’’. The search strategy for the MEDLINE (Ovid) database is available in the Appendix A of this manuscript.

### 2.2. Inclusion Criteria

The inclusion criteria were full-length research articles published in peer-reviewed journals in the English language with no limits set on the date of first publication or gender. Only RCTs up to January 2021 were eligible. Studies included participants who were diagnosed with CVD, such as those involved in some exercise programs, assessed by analyzing expired air during a maximal cardiopulmonary exercise test at baseline and postintervention.

We included RCTs to compare aerobic capacity changes resulting from exercise interventions, with an exercise group (or groups), that described exercise intensities, including data for risk ratio and CI.

Studies were required to detail the exercise prescription in patients with CVD, including the frequency, intensity and duration of each session, mode of exercise and the overall length of intervention. The main authors of studies and experts in this field were asked for any missed, unreported, or ongoing trials. The quantitative synthesis included studies reporting sample size and the mean and standard deviations (SDs) for VO_2_peak preintervention and postintervention.

### 2.3. Exclusion Criteria

Abstracts, conference presentations or posters, letters to editors or book chapters, unpublished papers, and retrospective design studies were excluded. In addition, studies were excluded if participants had documented heart failure (ejection fraction < 40%) or arrhythmia, they were targeting a specific comorbidity (e.g., diabetes, chronic obstructive pulmonary disease, or stroke) and they featured interventions involving resistance exercises only. We also excluded studies based on exercise prescriptions including testing food supplements and nutritional or pharmacological aids.

Studies were also excluded if baseline or postintervention data were not published, and the authors were not available for contact or did not wish to provide the missing data.

### 2.4. Study Selection and Data Extraction

All data were extracted by the principal investigator and their accuracy was assessed by the second author. The EndNote software (Clarivate Analytics, Philadelphia, PA, USA) was used to import, manage and remove duplicated articles for final review. After removing the duplicates, the two reviewers independently reviewed titles and abstracts against the inclusion/exclusion criteria. If in doubt, the full texts were evaluated to verify if they met the criteria. Subsequently, abstracts were selected for eligibility, and full manuscripts were retrieved for further evaluation of eligibility. Discrepancies were resolved between both authors, and a third expert, not involved in the previous procedures, was consulted to verify the ratings. The selection process was entered into a Preferred Reporting Items for Systematic Reviews and Meta-analysis (PRISMA) diagram [27] (Figure 1).

For each RCT, the author, year of publication, participant characteristics (age, gender, and primary diagnosis), description of the exercise testing protocol and description of the intervention (session frequency and duration, intervention length, exercise modality, resistance training, type of training (interval/continuous), supervision (clinic/home) and intervention type) were extracted. The pre- and post-VO_2_peak values and change in VO_2_peak were also extracted to assess change in CRF. Outcomes were extracted in relative (mL-kg^−1^-min^−1^) and absolute (L-min^−1^) terms. Outcomes reported in METs were converted to relative terms (METs × 3.5 mL-kg^−1^-min^−1^).

### 2.5. Assessment of Potential Bias

The risk of bias was assessed using the modified Cochrane collaboration tool [25], developed in 2005 to assess and report the risk of bias in RCTs. Bias assessment results from the judgment (high, low, or unclear) of individual elements from seven sources of bias covered six domains: random sequence generation (selection bias), allocation concealment (selection bias), blinding of participants and personnel (performance bias), blinding of outcome assessment (detection bias), incomplete outcome data (attrition bias), selective reporting (reporting bias) and other bias (criteria for selected patients in the studies and the country in which the study was conducted). A detailed description of each source of bias and support for judgement is available elsewhere [25]. The lead reviewer found 16 studies, and discrepancies were discussed and resolved.

### 2.6. Data Treatment and Analysis

The systematic review was stratified by intensities based on proposed cut-offs [28]. Thereby, each exercise program was ranked as light-, moderate- or vigorous-intensity aerobic exercise (Table 1).

Studies reporting an intensity that covers the categories of moderate intensity and vigorous intensity (e.g., 60–70% of VO_2_peak) were classified as “moderate-to-vigorous” intensity [28]. A separate meta-analysis was performed for each intensity category and length of the trial—e.g., “short-term” (0–6 weeks), “medium-term” (7–12 weeks), and “long-term” (>12 weeks).

The following subgroup analysis was conducted to explore significant heterogeneity: participant characteristics, including (1) age, (2) gender and (3) primary diagnosis; description of the exercise testing protocol and description of the intervention, including (4) session frequency and (5) duration, (6) intervention length, (7) exercise modality, (8) resistance training, (9) type of training (interval/continuous), (10) supervision (clinic/home), (11) intervention type (exercise only/comprehensive); and (12) pre- and postpeak VO_2_ values or change in VO_2_peak.

Heterogeneity amongst the included studies was first explored qualitatively by comparing the characteristics of the included trials and then by visually inspecting forest plots. It was also assessed quantitatively by the Chi^2^ and I^2^ statistics. Heterogeneity was considered minimal if I^2^ fell between 0–30%, moderate if 30–50%, substantial if 50–90%, and considerable if >90% [25]. I^2^ and Chi^2^ were considered significant at *p* < 0.1.

Due to the heterogeneity of the protocol, mean differences (MDs) were used, dividing the mean values between different intensities. The differences in means were grouped using the random-effects model. A random-effects model and a standardized means model of averages were used to explain the differences in the methodology of the studies included both in the intensities and length of intervention to ensure a conservative estimate was calculated. A sensitivity analysis was conducted to investigate the possible effects of specific studies on heterogeneity and overall effect.

The dichotomous and continuous variables of the studies were compared with the extracted potential VO_2_peak moderator factors. The effect of treatment was calculated for each study for the change in VO_2_peak over the intervention using the pooled between-subject SD at both time points. Effects were quantified as trivial (<0.20), small (0.21–0.60), moderate (0.61–1.20), large (1.21–2.00) and very large (>2.00) [29], with the precision of effect size estimates assessed using 95% CI. Pooled SMD was back-transformed using the pooled between-subject SD at baseline within each intensity category. If SD for the mean change in VO_2_peak across the intervention was not published [30], it was used for *p*-value entry. If no *p*-values or standard deviations were published, the standard error (SE) of the MD was inputted based on the correlation between preintervention and postintervention outcomes [31]. The imputed SE was then used to calculate the 95% CI for the standardized effect of each study. For outcomes expressed as change in relative VO_2_peak (mL-kg^−1^-min^−1^), a correlation of r = 0.54 from a similar meta-analysis [32] was used. A sensitivity analysis was performed using the estimated correlations of r = 0.30 and 0.70.

Publication bias was analyzed using a funnel plot derived in RevMan5.3 software [30]. The publication bias for the different conditions analyzed (pre- vs. postintervention) was assessed by examining the asymmetry of a funnel plot using Egger’s test, and *p* ≤ 0.05 was considered to be statistically significant.

## 3. Results

The initial search resulted in 221 studies. All data were extracted by the principal investigator and their accuracy was assessed by a second author. Search results were entered into EndNote software (Clarivate Analytics, Philadelphia, PA, USA), a reference management tool, and duplicates were removed. After the duplicates were removed, the titles of 212 studies were reviewed. Following a screening of potential records, 49 articles were reviewed for eligibility and their reference lists screened. Twenty-two RCTs met eligibility criteria for the systematic review and meta-analysis. According to our inclusion criteria, sixteen studies [9,10,11,12,13,14,20,21,22,33,34,35,36,37,38,39] were included in this systematic review (Figure 1).

The main characteristics of the studies and training interventions are described in Table 2 and Table 3, respectively.

### 3.1. Risk of Bias

Sixteen studies were scored by two reviewers, and an absolute agreement (r = 0.94) was obtained from the intraclass correlation coefficient (ICC). Bias was assessed as a judgment (high, low, or unclear) for individual elements from seven sources of bias and the following ICCs for absolute agreement between the two reviewers were obtained: random sequence generation for selection bias (r = 0.90), allocation concealment for selection bias (r = 0.92), blinding of participants and personnel for performance bias (r = 0.98), blinding of outcome assessment for detection bias (r = 0.94), incomplete outcome data for attrition bias (r = 0.79), selective reporting for reporting bias (r = 0.98) and inclusion criteria of patients in the studies and the country in which the study was conducted for other bias (r = 0.88). The risk of bias in the 16 included trials is summarized in Figure 2.

Of the 16 studies, the risk of bias was low in four or more of the seven sources of bias. Many studies were attributed to high risk in random sequence generation, allocation concealment and blinding of outcome assessment due to the nature of the exercise program. It was high in almost all studies due to the lack of blinding of participants and personnel. However, this issue could not be omitted due to the peculiarity of the intervention (exercise vs. no exercise) and should be taken into consideration.

The most prevalent methodological issues were an inadequate description of randomization (60%), allocation of concealment (50%) and blinding of outcome assessment (70%). Most studies were low risk for incomplete outcome data (90%).

### 3.2. Study and Participant Characteristics

The total number of CVD participants analyzed across all studies was 969 (267 coronary artery disease (CAD) only, 200 coronary artery bypass graft (CABG) only, 75 percutaneous coronary intervention (PCI) only, 50 CABG/PCI, 310 myocardial infarction (MI), and 63 carotid artery stiffness (CAS)). A summary of study characteristics is shown in the Appendix A.

The number of participants per group ranged between 15 and 48, with four studies reporting <20 participants and twelve studies reporting ≥20 participants, with the majority being males (*n* = 419). The age range of participants was 52–69 years, with nine studies reporting mean ages <60 years, six studies reporting mean ages ≥60 years, and one that did not report any age information. Individual patient characteristics for each study can be seen in Table 2.

Regarding the characteristics of the patients, the meta-analysis identified statistically significant improvements in VO_2_peak in each subgroup of patients with PCI (*p* < 0.001), as well as in patients with MI (*p* < 0.01), CABG (*p* < 0.02) and both CABG/PCI (*p* < 0.03).

### 3.3. Intervention Characteristics

The included trials tested a variety of interventions to increase VO_2_peak (Table 3). In many trials, the interventions were performed with exercise-based clinical supervision [10,11,12,13,14,20,21,22,34,37,38], a few studies implemented an unsupervised home-based program [33], and some studies performed both programs [9,30,36].

Exercise training was typically continuous [9,10,11,12,21,22,33,34,35,36,37,38,39], as opposed to interval [13,14], or mixed training [20], and this type of training was shown to be significantly superior in improving VO_2_peak (3.27 mL-kg^−1^-min^−1^; 95% CI = 2.23–4.32; *p* < 0.001; I^2^ = 87%).

The frequency of training was typically 3–4 days/week [9,10,11,12,13,14,20,21,22,33,34,35,38], and aerobic training was the most used type of intervention [10,12,13,14,20,21,22,33,34,35,36,38,39]. Three studies tested aerobic and resistance training together during the intervention [9,11,37]. The meta-analysis identified that cycle-ergometers (*p* < 0.05) and treadmill (*p* < 0.01) significantly favored changes in VO_2_peak.

Studies were separated into three groups depending upon length (<six, 6–12, and >12 weeks). The intervention length ranged from two to 24 weeks, with one study that reported data for less than six weeks [37], nine studies reported data for 6 to 12 weeks [9,10,13,20,31,33,34,36,37], and six studies reported data for >12 weeks [11,12,14,21,22,35]. The subgroup that included studies of >12 weeks in length was significantly superior in terms of improvements in VO_2_peak (2.50 mL-kg^−1^-min^−1^; 95% CI = 2.23–4.32; *p* < 0.001; I^2^ = 52%). Interventions 6 to 12 weeks in length also produced a large increase (*p* < 0.01), demonstrating moderate heterogeneity (5.31 mL-kg^−1^-min^−1^; 95% CI = 1.24–9.38; I^2^ = 52%).

Based on the American College of Sports Medicine (ACSM) [28] cut-off points, three studies prescribed moderate-intensity exercise (*n* = 18, 75%) [10,33,38], three prescribed vigorous-intensity exercise (*n* = 18, 75%) [14,20,21] and ten interventions (*n* = 62,5%) [9,11,13,22,35,36,37,39] prescribed a range of intensities that placed them within both the moderate-intensity and vigorous-intensity categories. The meta-analyzed effects found the intervention was beneficial in terms of changing VO_2_peak in both intensities (*p* < 0.001).

### 3.4. Subgroup Analyses—Intensity

When interpreting these results (Figure 3), it is essential to consider how exercise intensity was classified. We used a categorical-based approach, in which interventions were categorized according to the prescribed exercise intensity reported in each study, based on the recommendations of the ACSM [28].

The meta-regression analysis displayed in Figure 3 revealed that relative VO_2_peak was significantly increased in all intensity categories. Moderate-intensity interventions produced a moderate increase in relative VO_2_peak (0.71 mL-kg^−1^-min^−1^; 95% CI = 0.27–1.15; *p* = 0.001) with moderate heterogeneity (I^2^ = 45%). Moderate-to-vigorous-intensity and vigorous-intensity interventions produced a large increase in relative VO_2_peak (1.84 mL-kg^−1^-min^−1^; 95% CI = 1.18–2.50; *p* < 0.001 and 1.80 mL-kg^−1^-min^−1^; 95% CI = 0.82–2.78; *p* = 0.0003, respectively), and were also highly heterogeneous with I^2^ values of 91 and 95% (*p* < 0.001), respectively.

### 3.5. Subgroup Analyses—Intensity and Length

In the analyses of studies lasting less than six weeks, we evaluated studies that exercised at vigorous intensity. The results (Figure 4) showed a large increase in VO_2_peak (3.81 mL-kg^−1^-min^−1^; 95% CI = 0.16–7.45; *p* = 0.04) and demonstrated significant heterogeneity (I^2^ = 98%).

For interventions of 6 to 12 weeks length, moderate-to-vigorous-intensity interventions showed a further increase in VO_2_peak (2.28 mL-kg^−1^-min^−1^; 95% CI = 1.23–3.32; *p* < 0.001; I^2^ = 93%) compared to moderate-intensity (0.71 mL-kg^−1^-min^−1^; 95% CI = 0.12–1.29; *p* = 0.02; I^2^ = 0%) and vigorous-intensity interventions (1.57 mL-kg^−1^-min^−1^; 95% CI = 0.12–3.02; *p* = 0.02; I^2^ = 0%).

For studies that intervened more than 12 weeks, moderate-to-vigorous-intensity interventions were significantly superior (1.07 mL-kg^−1^-min^−1^; 95% CI = 0.64–1.50; *p* < 0.001; I^2^ = 51%) to vigorous-intensity interventions (0.92 mL-kg^−1^-min^−1^; 95% CI = −0.12–1.96; *p* = 0.08; I^2^ = 82%) in improving VO_2_peak.

### 3.6. Publication Bias

There was no significant publication bias for studies with moderate-intensity (Egger’s test: *β* = 7.29; *p* = 0.26) and vigorous-intensity (Egger’s test: *β* = 8.67; *p* = 0.15) interventions reporting relative VO_2_peak. However, there was significant publication bias for studies with moderate-to-vigorous-intensity interventions (Egger’s test: *β* = 13.19; *p* = 0.00). The funnel plot with all studies (Figure 5) showed a significant degree of asymmetry (Egger’s test: *p* = 0.00). Nevertheless, false-positive results may occur due to substantial between-study heterogeneity [40], making the disparity in the number of studies included in each intensity category likely to cause significant asymmetry in the funnel plot.

## 4. Discussion

The main aim of this systematic review and meta-analysis was to identify the optimal intensity to optimize VO_2_peak in patients with CVD following exercise programs. Furthermore, we aimed to gauge whether the length of interventions had an effect on the results.

Our results support the crucial role of physical exercise in patients with CVD. They have shown significant improvements for all cardiac impairments at all ages, regardless of the aerobic exercise mode.

A comparison of the mean effects between intensity classifications showed significant improvements, with moderate-to-vigorous-intensity interventions providing the greatest improvements of VO_2_peak. The differences were considered clinically significant (*p* = 0.03) and the retro transformation of the SMD suggested that the difference between the intensities was 3.92 mL-kg^−1^-min^−1^. However, when comparing the effects grouped among the intensity classifications, it was found that moderate-to-vigorous-intensity exercises can provide the most significant improvements in VO_2_peak. Even so, the differences were not considered clinically significant once the retro transformation of the SMD suggested that the differences between the intensities were, at most, only 1.67 mL-kg^−1^-min^−1^. In this regard, our study confirmed the results of previous systematic reviews, pointing out that moderate-to-vigorous- and vigorous-intensity interventions improved CRF to a larger extent than moderate-intensity ones [23].

The difference between moderate-to-vigorous- and vigorous-intensity in our study was 0.4 mL-kg^−1^-min^−1^ and the difference between moderate- and moderate-to-vigorous-intensity was more significant (1.13 mL-kg^−1^-min^−1^). Although these analyses did not yield any consistent findings, they highlighted considerable variability in outcomes for interventions based on VO_2_peak that appeared to be consistent across intensities. Although unexpected, this finding is not surprising. Given that VO_2_ is not an appropriate variable to regulate intensity during training, in practice, prescriptions are converted to heart rate (HR) estimated to elicit the target VO_2_. This approach is confounded in a CR setting by medications (e.g., β-blockers) that alter HR responses, which may cause dissociation of the HR and VO_2_ relationship, where a small change in HR may result in varied and disproportionate changes to work rate or VO_2_peak [8,17,22].

The first meta-analyses that investigated improvements in CRF following exercise-based CR reported a small improvement in CRF (SMD ±: 95% CI = 0.46 ± 0.02) [41]. Our study confirmed the results of Mitchell et al. [23] who verified that moderate- and moderate-to-vigorous-intensity interventions were associated with a moderate increase in relative VO_2_peak (SMD ±: 95% CI = 0.94 ± 0.30 and 0.93 ± 0.17, respectively), and vigorous-intensity exercise with a large increase (SMD ±: 95% CI = 1.10 ± 0.25), and moderate- and vigorous-intensity interventions were associated with moderate improvements in absolute VO_2_peak (SMD ±: 95% CI = 0.63 ± 0.34 and SMD ±: 95% CI = 0.93 ± 0.20, respectively), whereas moderate-to-vigorous- intensity interventions elicited a large effect (SMD ±: 95% CI = 1.27 ± 0.75).

When we subdivided the intensities by length to obtain a more in-depth view of the effect of the different intensities, we found that the vigorous-intensity interventions below six weeks had more significant results in improving the VO_2_peak (3.81 mL-kg^−1^-min^−1^). Based on the sensitivity analysis, although the results suggest that interventions conducted bidirectionally six times a week resulted in more significant gains of CRF favoring vigorous intensity, the analysis only included two studies and may not be practical to implement. In this sense, not being able to compare with other studies and other intensities within the division by length, the best result obtained was between 6 and 12 weeks in moderate-to-vigorous-intensity exercise, in which there was a significant increase in VO_2_peak in relation to the vigorous- and moderate-intensity categories.

Interventions >12 weeks did not show significantly greater gains in CRF compared to other lengths. However, there was a significant improvement in VO_2_peak with moderate-to-vigorous intensity. Additionally, there was no significant improvement in VO_2_peak with vigorous-intensity interventions, and there were no studies of moderate-intensity RCTs available for comparison. Furthermore, patients with CVD did not obtain significant VO_2_peak improvements when the vigorous-intensity protocol was >12 weeks.

Our results indicate that moderate-to-vigorous-intensity exercise is superior to other intensities in improving aerobic capacity and is likely to be an underestimation of the true differences between groups. This is supported by the methodological decisions favoring the use of a conservative approach in the meta-analysis (by choosing random effects and SMDs) and using the highest calculated SD for studies where no information was published to allow SD calculations.

Thereby, our findings suggest higher benefits from moderate-to-vigorous-intensity exercise lasting 6 to 12 weeks in terms of VO_2_peak improvements in patients with CVD. Overall, our findings are in agreement with reports from previous meta-analyses [19,23,32,42,43]. Hannan et al. [24] concluded that HIIT (e.g., of moderate-to-vigorous- and vigorous intensity) is more effective than moderate-intensity exercise in improving CRF in participants of CR (0.34 mL-kg^−1^-min^−1^; 95% CI = 0.2–0.48; *p* < 0.001; I^2^ = 28%). Still, improvements in CRF were higher in >six-week exercise programs, and the largest improvements in CRF for patients with CAD resulted from programs lasting 7 to 12 weeks, as our study confirmed [24].

Some limitations of this systematic review and meta-analysis should be considered. First, the poor level of reporting within the available RCTs made it difficult to evaluate the most effective doses of intensity on CRF in cardiac patients. Second, the RCTs did not use the same methods to control the exercise intensity and the different variables used to establish exercise intensity added complexity to the analyses. While the variables were based on interrelated physiological constructs (e.g., HR and VO_2_), they were not directly comparable. Even in what appears to be the narrow domain of HIIT, there is much heterogeneity in clearly defining what high intensity is.

In our study, each reported intervention was categorized according to the prescribed exercise intensity, based on ACSM recommendations [28]. This approach has two limitations. While some studies reported precise exercise intensities (e.g., 60% VO_2_peak), most CR studies prescribed large intervals based on HR responses to exercise (e.g., 40–70% VO_2_peak). As these studies often covered several intensity categories, making them difficult to categorize, it was necessary to add an extra intensity category, moderate-to-vigorous intensity. In this category, participants were assumed to have performed similar training interventions, when in fact they may have experienced quite different exercise prescriptions.

We recognized the lack of available data for some intensity analyses when split by program length. For example, in the analysis of subgroups of studies below six weeks, we only had studies that prescribed vigorous-intensity exercise, as well as in the length above 12 weeks, we had no studies that used in their intervention a moderate-intensity program. Furthermore, subgroup analyses for the combined effect of vigorous-intensity programs with lengths below six weeks were based on only two groups of patients, both from the same study that completed the same intervention. As such, we recommend caution when interpreting results where the lack of available data may have limited analyses.

We should consider that medication can influence exercise and therefore should be considered by the therapist when prescribing exercise. Beta-blockers decrease exercise capacity because they create a ceiling effect, meaning the HR will not rise beyond a certain point. Thus, the target HR for monitoring should not be used. Rather, the therapist should use the rate of perceived exertion or calculate the target HR with a graded stress test while the patient is using the medication. Similarly, vasodilators and alpha- and calcium channel blockers may lead to a sudden blood pressure drop while exercising or afterwards.

Therefore, the variables that should be taken into consideration are trainability (result of CRF level, muscular endurance and strength) and risk stratification on the basis of completed medical history. Consequently, these factors may provide options for the optimal type of exercise and intensity level.

Future studies would benefit from being between 6 and 12 weeks in length with an intervention activity carried out at least three times weekly, ensuring that the correct intensity is maintained. For example, appropriate goals for vigorous-intensity exercise include ≥85% VO_2_peak or ≥85% HRR or ≥90% HRM and, for moderate intensity, 50–75% VO_2_peak or 50–75% HRR or 50–80% HRM. In addition, large ranges of exercise intensities should not be prescribed based on HR responses to exercise. This would allow a more accurate calculation of the exact effects of intensities on CRF and to determine the ideal and most effective “dose” for people with heart problems. Future research should include methods to appropriately describe the compliance of participants with the prescribed exercise intensity and attendance of exercise sessions.

Studies should report standard deviations, conceal allocation, and blind assessors to improve study quality. Moreover, future studies should aim to recruit more women and older participants (<76 years) to ensure vigorous-intensity interventions are more effective than moderate-intensity ones in improving CRF for a broader range of patients with CVD. Finally, further studies that investigate the longer-term benefits of vigorous-intensity interventions and whether these adaptations are maintained would also be beneficial.

## 5. Conclusions

The most effective doses of exercise intensity to optimize CRF were moderate-to-vigorous and vigorous exercise. Interventions to enhance CRF in patients with CVD are most effective if conducted for 6 to 12 weeks. More research is needed to understand within the moderate-to-vigorous-intensity category which percentage results in increased CRF, assisting in the design of specific prescription protocols.

This review may suggest that countries without guidelines for patients with CVD regarding the intensity of exercise programs, as well as countries with guidelines that recommend lower intensity exercise, should include moderate-to-vigorous intensity and vigorous intensity.

What is already known:►Cardiovascular diseases are the leading causes of mortality in today’s society. They are responsible for up to 30% of all deaths worldwide and 48% of deaths in Europe, and it is expected that these figures will increase in the coming years.►Exercise programs in patients with cardiovascular disease have several beneficial effects on cardiovascular functional capacity, quality of life, risk factors modification, psychological profile, hospital readmissions, and mortality.►Exercise-based interventions seem to significantly improve cardiorespiratory fitness in patients following a cardiac event or surgery, but little is known regarding the differential effects of prescribed exercise intensity.

What are the new findings?

►Exercise interventions for patients with cardiovascular disease tend include large ranges of exercise intensities based on heart rate responses to exercise.►The most effective doses of exercise intensity to optimize cardiorespiratory fitness were moderate-to-vigorous and vigorous-intensity exercises, being more effective when conducted for 6 to 12 weeks.►More research is needed to understand within the moderate-to-vigorous- and vigorous-intensity categories the percentage that specifically helps to increase cardiorespiratory fitness and the ability to establish specific prescription protocols.

## Figures and Tables

**Figure 1 ijerph-18-03574-f001:**
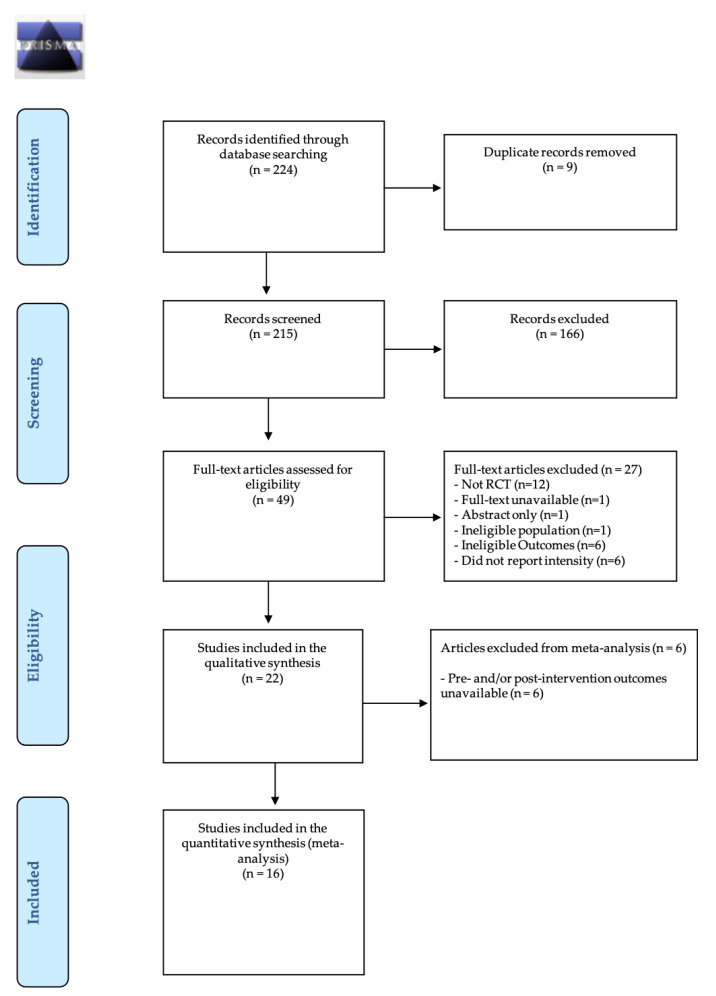
Preferred Reporting Items for Systematic Reviews and Meta-analysis (PRISMA) diagram of literature search strategies.

**Figure 2 ijerph-18-03574-f002:**
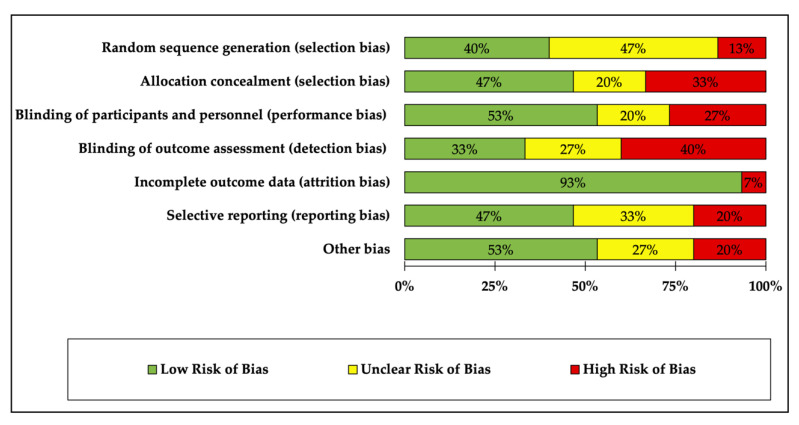
Assessment of risk of bias in included randomized controlled trials.

**Figure 3 ijerph-18-03574-f003:**
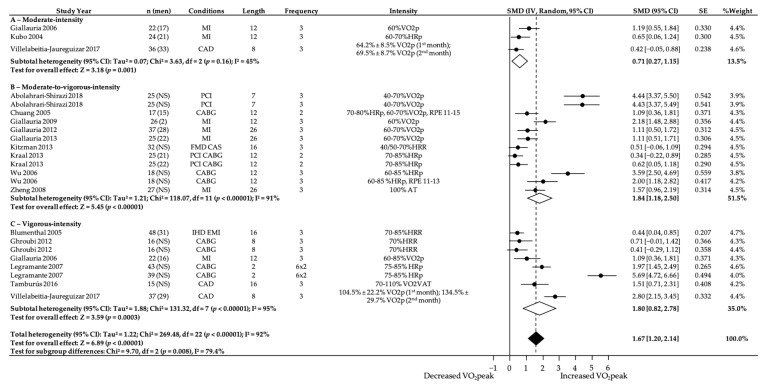
Effect of moderate-, moderate-to-vigorous- and vigorous-intensity exercise during exercise programs on change in relative VO_2_peak (mL-kg^−1^-min^−1^). NS, not stated/missing. HRR, heart rate reserve. HRp, heat rate peak. RPE, rate of perceived exertion. AT, anaerobic threshold. VAT, ventilatory anaerobic threshold. 95% CI, 95% confidence interval. SMD, standardized mean difference. IV, Random: a random-effects meta-analysis was applied, with weights based on inverse variances. SE, standard error. Tau^2^ and I^2^, heterogeneity statistics. df, degree of freedom. Chi^2^, the chi-squared test value. Z, Z-value for test of the overall effect. P, p-value. Conditions: MI, myocardial infarction. CABG, coronary artery bypass graft. PCI, percutaneous coronary intervention. CAD, coronary artery disease. IHD, ischsemic heart disease. EMI, exercise-induced myocardial ischemia. FMD, endothelial-dependent flow-mediated arterial dilation. CAS, carotid artery stiffness.

**Figure 4 ijerph-18-03574-f004:**
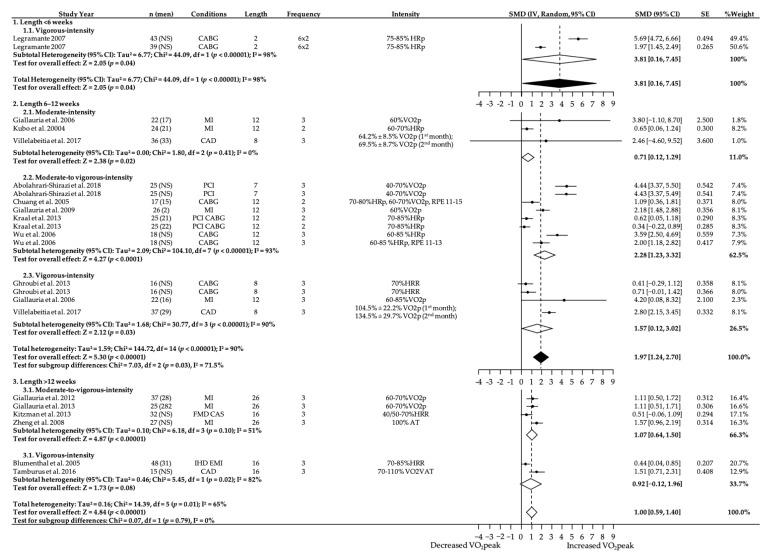
Effect of length in moderate-, moderate-to-vigorous- and vigorous-intensity exercise during exercise programs on change in relative VO_2_peak (mL-kg^−1^-min^−1^). NS, not stated/missing. HRR, heart rate reserve. HRp, heat rate peak. RPE, rate of perceived exertion. AT, anaerobic threshold. VAT, ventilatory anaerobic threshold. 95% CI, 95% confidence interval. SMD, standardized mean differences. IV, Random: a random-effects meta-analysis is applied, with weights based on inverse variances. SE, standard error. Tau^2^ and I^2^, heterogeneity statistics. df, degrees of freedom. Chi^2^, the chi-squared test value. Z, Z-value for test of the overall effect. P, p-value. Conditions: MI, myocardial infarction. CABG, coronary artery bypass graft. PCI, percutaneous coronary intervention. CAD, coronary artery disease. IHD, ischemic heart disease. EMI, exercise-induced myocardial ischemia. FMD, endothelial-dependent flow-mediated arterial dilation. CAS, carotid artery stiffness.

**Figure 5 ijerph-18-03574-f005:**
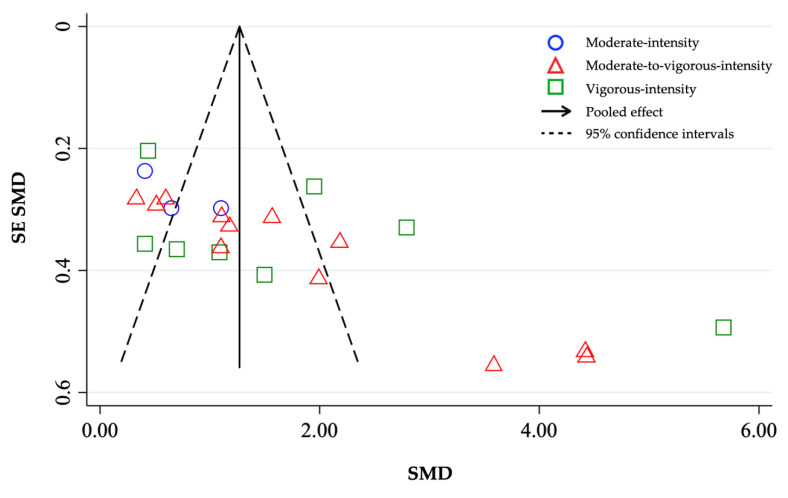
Funnel plot with pseudo 95% confidence intervals for change in relative VO_2_peak (mL-kg^−1^-min^−1^) by exercise intensity (moderate, moderate-to-vigorous, vigorous). SMD, standardized mean difference. SE SMD, standard error of standardized mean differences.

**Table 1 ijerph-18-03574-t001:** Classification of exercise intensity based on physiological and perceived exertion responses.

	%VO_2max_	%HR_peak_	%HR_reserve_/%VO_2reserve_	Perceived Exertion *
Light	37–45	57–63	30–39	RPE 9–11
Moderate	46–63	64–76	40–59	RPE 12–13
Vigorous	64–90	77–95	60–89	RPE 14–17
Near maximal to maximal	≥91	≥95	≥90	RPE ≥ 18

Table adapted from American College of Sports Medicine (ACSM) [28] and Mitchell et al. [23]. * As per the Borg 6–20 RPE scale. %VO_2_max, percentage of maximal oxygen uptake; %HRpeak, percentage of peak heart rate; %HRreserve, percentage of heart rate reserve; %VO_2_reserve, percentage of oxygen uptake reserve; RPE, rating of perceived exertion.

**Table 2 ijerph-18-03574-t002:** Subgroup analyses assessing potential moderating factors for VO_2_peak increase in studies included in the meta-analysis by population characteristics.

Research Studies			Peak VO_2_			
Group	N	References	MD (95% CI)	I^2^	*p* ^a^	*p*-Difference ^b^
No. of participants						
<20	4	Ghroubi et al. [20], Tamburus et al. [14], Wu et al. [33], Chuang et al. [34]	2.62 (1.65, 3.58)	88	<0.001	0.78
≥20	12	Abolahrari-Shirazi et al. [9], Blumenthal et al. [21], Giallauria et al. [10,11,12,36], Kitzman et al. [22], Kraal et al. [36], Kubo et al. [38], Legramante et al. [37], Villelabeitia et al. [13], Zheng et al. [35]	2.75 (2.58, 2.93)	97	<0.001	
Age, years						
<60	9	Abolahrari-Shirazi et al. [9], Ghroubi et al. [20], Giallauria et al. [10,12,36], Kraal et al. [39], Kubo et al. [38], Tamburus et al. [14], Villelabeitia et al. [13]	4.40 (0.79, 8.01)	97	0.02	0.75
≥60	6	Blumenthal et al. [21], Chuang et al. [34], Giallauria et al. [11], Kitzman et al. [22], Legramante et al. [37], Wu et al. [33]	3.48 (2.09, 4.87)	79	<0.001	
Not reported	1	Zheng et al. [35]	3.10 (2.06, 4.14)	0	<0.001	
Diagnosis						
CAD only	3	Blumenthal et al. [21], Tamburus et al. [14], Villelabeitia et al. [13]	6.41 (−2.70, 15.53)	99	0.17	0.03
CABG only	4	Chuang et al. [34], Ghroubi et al. [20], Legramante et al. [37], Wu et al. [33]	4.27 (1.60, 6.94)	85	0.002	
PCI only	1	Abolahrari-Shirazi et al. [9]	8.20 (4.68, 11.72)	0	<0.001	
CABG/PCI	1	Kraal et al. [39]	3.20 (0.36, 6.04)	0	0.03	
MI	6	Giallauria et al. [10,11,12,36], Kubo et al. [38], Zheng et al. [35]	2.65 (0.56, 4.74)	91	0.01	
FMD	1	Kitzman et al. [22]	1.60 (−0.13, 3.33)	0	0.07	
Study location						
America	2	Kitzman et al. [22], Tamburus et al. [14]	1.38 (0.39, 2.36)	0	0.006	0.01
Africa	1	Ghroubi et al. [20]	1.70 (−1.07, 4.47)	0	0.23	
Asia	5	Abolahrari-Shirazi et al. [9], Chuang et al. [34], Kubo et al. [38], Wu et al. [33], Zheng et al. [35]	5.33 (2.90, 7.76)	80	<0.001	
Europe	8	Blumenthal et al. [21], Giallauria et al. [10,11,12,36], Kraal et al. [39], Legramante et al. [37], Villelabeitia et al. [13]	4.23 (1.50, 6.95)	98	0.002	

95% CI, 95% confidence interval. I2, heterogeneity. MD, mean difference. Peak VO2, peak oxygen uptake. Conditions: MI, myocardial infarction. CABG, coronary artery bypass graft. PCI, percutaneous coronary intervention. CAD, coronary artery disease. FMD, endothelial-dependent flow-mediated arterial dilation. Certain enrolled studies were not included because the value used for subgroup analysis was not reported in them. a Test for overall effect. b Test for subgroup differences.

**Table 3 ijerph-18-03574-t003:** Subgroup analyses assessing potential moderating factors for VO_2_peak increase in studies included in the meta-analysis by population characteristics.

Research Studies			Peak VO_2_				
Group	N	References	MD (95% CI)	I^2^	*p* ^a^	*p*-Difference ^b^	
Length, weeks							
<6	1	Legramante et al. [37]	2.60 (2.41, 2.79)	0	<0.001	0.42	
6–12	9	Abolahrari-Shirazi et al. [9], Chuang et al. [34], Ghroubi et al. [20], Giallauria et al. [10,36], Kraal et al. [39], Kubo et al. [38], Villelabeitia et al. [13], Wu et al. [33]	5.31 (1.24, 9.38)	97	0.01		
>12	6	Blumenthal et al. [21], Giallauria et al. [11,12], Kitzman et al. [22], Tamburus et al. [14], Zheng et al. [35]	2.50 (1.60, 3.41)	52	<0.001		
Frequency, sessions/week							
1–2	2	Chuang et al. [34], Kraal et al. [39]	3.98 (1.96, 6.01)	0	0.001	0.17	
3–4	13	Abolahrari-Shirazi et al. [9], Blumenthal et al. [21], Ghroubi et al. [20], Giallauria et al. [10,11,12,36], Kitzman et al. [22], Kubo et al. [38], Tamburus et al. [14], Villelabeitia et al. [13], Wu et al. [33], Zheng et al. [35]	4.21 (1.82, 6.60)	96	0.006		
5–7	1	Legramante et al. [37]	2.60 (2.41, 2.79)	0	<0.001		
Supervision							
Clinic	12	Blumenthal et al. [21], Chuang et al. [34], Ghroubi et al. [20], Giallauria et al. [10,11,12], Kitzman et al. [22], Kubo et al. [38], Legramante et al. [37], Tamburus et al. [14], Villelabeitia et al. [13], Zheng et al. [35]	4.01 (2.30, 5.72)	96	<0.001	0.02	
Home	1	Wu et al. [33]	8.50 (5.78, 11.22)	0	<0.001		
Mixed	3	Abolahrari-Shirazi et al. [9], Giallauria et al. [36], Kraal et al. [39]	2.99 (−2.89, 8.87)	94	0.32		
Intervention type							
Continuous	13	Abolahrari-Shirazi et al. [9], Blumenthal et al. [21], Chuang et al. [34], Giallauria et al. [11,12,36], Kitzman et al. [22], Kraal et al. [39], Kubo et al. [38], Legramante et al. [37], Wu et al. [33], Zheng et al. [35]	3.27 (2.23, 4.32)	87	<0.001	0.44	
Interval	2	Tamburus et al. [14], Villelabeitia et al. [13]	8.67 (−5.86, 23.21)	99	0.24		
Mixed	1	Ghroubi et al. [20]	1.70 (−1.07, 4.47)	0	0.23		
Mode							
Cycle ergometer	7	Ghroubi et al. [20], Giallauria et al. [10,11,12], Tamburus et al. [14], Villelabeitia et al. [13], Zheng et al. [35]	4.90 [1.52, 8.27)	97	0.005	0.23	
Treadmill	1	Chuang et al. [34]	4.80 (1.91, 7.69)	0	0.001		
Walking	1	Blumenthal et al. [21]	1.90 (0.20, 3.60)	0	0.03		
Mixed (treadmill, walking, cycling, calisthenics or/and arm/leg ergometer)	7	Abolahrari-Shirazi et al. [9], Giallauria et al. [36], Kitzman et al. [22], Kraal et al. [39], Kubo et al. [37], Legramante et al. [37], Wu et al. [33]	3.28 (1.17, 5.39)	92	0.002		
Exercise type							
Aerobic	13	Blumenthal et al. [21], Chuang et al. [34], Ghroubi et al. [20], Giallauria et al. [10,12,36], Kitzman et al. [22], Kraal et al. [39], Kubo et al. [38], Tamburus et al. [14], Villelabeitia et al. [13], Wu et al. [33], Zheng et al. [35]	3.94 (1.55, 6.34)	96	0.001	0.86	
Aerobic and Resistance	3	Abolahrari-Shirazi et al. [9], Giallauria et al. [11], Legramante et al. [37]	4.24 (1.82, 6.67)	81	0.001		
Intensity							
Moderate	3	Giallauria et al. [10], Kubo et al. [38], Villelabeitia et al. [13]	2.90 (1.64, 4.16)	0	<0.001	0.03	
Moderate-to-vigorous	10	Abolahrari-Shirazi et al. [9], Chuang et al. [34], Giallauria et al. [11,12,36], Kitzman et al. [22], Kraal et al. [39], Wu et al. [33], Zheng et al. [35]	5.07 (3.43, 6.72)	92	<0.001		
Vigorous	3	Blumenthal et al. [21], Ghroubi et al. [20], Giallauria et al. [10], Legramante et al. [37], Tamburus et al. [14], Villelabeitia et al. [13]	2.43 (1.33, 3.54)	75	<0.001		

95% CI, 95% confidence interval. I^2^, heterogeneity. MD, mean difference. Peak VO2, peak oxygen uptake. Certain enrolled studies were not included because the value used for subgroup analysis was not reported in them. ^a^ Test for overall effect. ^b^ Test for subgroup differences.

## Data Availability

The data that support the findings of this study are available from the corresponding author, C.G., upon reasonable request.

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
