# Peer review of "Exercise Intensity in Patients with Cardiovascular Diseases: Systematic Review with Meta-Analysis"

_ijerph, 2021, doi:10.3390/ijerph18073574_

Round 1

Reviewer 1 Report

Congratulations to the authors. The manuscript presents interesting and very thorough research.
Some considerations are presented below for the authors to value and include if they consider them appropriate.

ABSTRACT

- It is recommended to include information on the data sources consulted.

INTRODUCTION

- Line 45. The acronym CRF appears for the first time but was not clarified earlier in the text.

- Line 84. The same occurs with the acronym Randomized Controlled Trials.

- Lines 86-88. This information should appear in detail in the corresponding sections of Material and Methods. "Utilizing RCT studies published up to January 2021, we present the data as the risk difference (95% CI), which ensures all studies reporting data on the outcomes of interest were included."

MATERIALS ANS METHODS

- Search strategy. It is advisable to present the complete electronic search strategy in at least one of the databases used, so that it can be reproducible.

- Study selection and data extraction. Lines 142-143. It seems that this information repeats what has already been explained in the previous paragraph and creates confusion.  Please review both paragraphs and clarify the process.

- Figure 1 is blurred.

- Assessment of potential bias. Describe in more detail the methods used to assess the risk of bias in the 7 domains mentioned.

- Data treatment and analysis. Indicate the values considered for Chi2.

RESULTS

- Table 2. There is a typo in the range of references "Abolahrari-Shirazi et al.[9], Blumenthal et al.[21], Giallauria et al.[12-12,33],"

- Line 249. Where it says "Table 4 (continued)" should read "Table 3 (continued).

- Figures 2, 3 and 4 are blurred.

- Intervention characteristics. Line 298. "(Error! Reference source not found.Table 3)" appears. Correct.

- Meta-regression. Relationships are mentioned, but their data are missing.

DISCUSSION

- Consider including the subsection "Application to practice" in the discussion (before or after the limitations) instead of in the conclusion.

PATENTS

- Line 548. The web address is incomplete (https://www.mdpi.com/???????). Correct.

Reviewer 2 Report

his systematic review with meta-analysis is an important topic for the health and Exercise specialists. However, this study lack of information in some points. Altogether, it is a well written manuscript.

L81-88. It would be better to link with the previous paragraph to raise the research gap. The authors may state the research gap based on the previous studies. Study XX assessed YY but did not evaluated ZZ. This will help to better understand the research gap. The Information is presented in the paragraphs. However, it is very fragmented.

Methods: this is a systematic review with meta-analysis. The authors have assessed potential bias, why the authors did not assessed the studies quality with an instrument?

Please improve Fig2 quality.

Results: figure 3 has low quality. Why not use a table and in the SMD column the authors paste the image? Same for fig4.

Figure 5: remove the title from the plot. Letters type are different from the manuscript.

Discussion: the authors may justify their methods in the second paragraph. Why PRISMA? Why Cochrane Collaboration for potential bias? Why did the authors not classify the studies with a scale such as PEDro or other?

Does this study have not limitations?

5.1. "Practical Application"?

List of references is incomplete. It stats in reference 9.

This reviwer believes that the manuscript, specially the discussion must have more references supporting the results of the included studies. The discussion is too much focused on the results.

Round 2

Reviewer 2 Report

The authors have adressed my points and I do believe that this manuscript should be considered for publication.

The authors well explained my questions.